# Structural and social determinants of health: The multi-ethnic study of atherosclerosis

Lilah M. Besser[1]*, Sarah N. Forrester[2], Milla Arabadjian[3], Michael P. Bancks[4], Margaret Culkin[5], Kathleen M. Hayden[5], Elaine T. Le[1], Isabelle Pierre-Louis[2], Jana A. Hirsch[6]

1 Department of Neurology, Comprehensive Center for Brain Health, University of Miami, Boca Raton, Florida, United States of America, 2 Division of Epidemiology, Department of Population and Quantitative Health Sciences, University of Massachusetts Chan Medical School, Worcester, Massachusetts, United States of America, 3 Department of Foundations of Medicine, NYU Grossman Long Island School of Medicine, Mineola, New York, United States of America, 4 Department of Epidemiology and Prevention, Wake Forest University School of Medicine, Winston-Salem, North Carolina, United States of America, 5 Department of Social Sciences and Health Policy, Wake Forest University School of Medicine, Winston-Salem, North Carolina, United States of America, 6 Urban Health Collaborative and Department of Epidemiology and Biostatistics, Dornsife School of Public Health, Drexel University, Philadelphia, Pennsylvania, United States of America

* lmb9767@miami.edu

**Data Availability Statement:** All relevant data are within the manuscript and its Supporting Information files.

**Funding:** The author(s) received no specific funding for this work.

## Abstract

### Background

Researchers have increasingly recognized the importance of structural and social determinants of health (SSDOH) as key drivers of a multitude of diseases and health outcomes. The Multi-Ethnic Study of Atherosclerosis (MESA) is an ongoing, longitudinal cohort study of subclinical cardiovascular disease (CVD) that has followed geographically and racially/ethnically diverse participants starting in 2000. Since its inception, MESA has incorporated numerous SSDOH assessments and instruments to study in relation to CVD and aging outcomes. In this paper, we describe the SSDOH data available in MESA, systematically review published papers using MESA that were focused on SSDOH and provide a roadmap for future SSDOH-related studies.

### Methods and findings

The study team reviewed all published papers using MESA data (n = 2,125) through January 23, 2023. Two individuals systematically reviewed titles, abstracts, and full text to determine the final number of papers (n = 431) that focused on at least one SSDOH variable as an exposure, outcome, or stratifying/effect modifier variable of main interest (discrepancies resolved by a third individual). Fifty-seven percent of the papers focused on racialized/ethnic groups or other macrosocial/structural factors (e.g., segregation), 16% focused on individual-level inequalities (e.g. income), 14% focused on the built environment (e.g., walking destinations), 10% focused on social context (e.g., neighborhood socioeconomic status), 34% focused on stressors (e.g., discrimination, air pollution), and 4% focused on social support/integration (e.g., social participation). Forty-seven (11%) of the papers combined MESA

**Competing interests:** The authors have declared that no competing interests exist.

with other cohorts for cross-cohort comparisons and replication/validation (e.g., validating algorithms).

## Conclusions

Overall, MESA has made significant contributions to the field and the published literature, with 20% of its published papers focused on SSDOH. Future SSDOH studies using MESA would benefit by using recently added instruments/data (e.g., early life educational quality), linking SSDOH to biomarkers to determine underlying causal mechanisms linking SSDOH to CVD and aging outcomes, and by focusing on intersectionality, understudied SSDOH (i.e., social support, social context), and understudied outcomes in relation to SSDOH (i.e., sleep, respiratory health, cognition/dementia).

## Introduction

Social determinants of health (SDOH) are the "conditions in the environments where people are born, live, learn, work, play, worship, and age that affect a wide range of health, functioning, and quality-of-life outcomes and risks" [1]. SDOH have been categorized and conceptualized using a wide range of conceptual frameworks, but in common, these frameworks outline how SDOH have multiple spheres of influence (e.g., individual, interpersonal, community, and societal levels) and encompass multiple domains (e.g., sociocultural environment and neighborhood physical/built environment) [2]. Numerous health conditions and behaviors have been associated with SDOH, including but not limited to physical activity, diet, cardiovascular and cerebrovascular disease, cognitive function and dementia risk, and mortality (e.g., [3–6]). Differences in numerous health outcomes have been observed by minoritized racial and ethnic groups, which are socially constructed identities that have experienced longstanding systemic racism that continues to impose on these groups lower access to socioeconomic and health-promoting opportunities and resources [7, 8]. Residents in zip codes with greater racial/ethnic segregation and lower socioeconomic status have shorter life expectancies. For instance, cities such as Chicago and Washington, D.C. experience significant gaps in life expectancy (i.e., 28 to 30 years difference) depending on the zip code of residence [9].

SDOH are influenced by the overarching structural determinants of health, which are the "social and political mechanisms that generate … stratification and social class divisions in society and that define individual socioeconomic position within hierarchies of power, prestige and access to resources" [10]. Considered together, structural and social determinants of health (SSDOH) are major causes of poorer health outcomes and disparities for economically disadvantaged, racialized, and minoritized individuals and communities [11]. Targeted interventions (e.g., policies, programs, new infrastructure) are needed to reduce socioeconomic and racialized/ethnic disparities in a wide range of health outcomes, but these interventions must be evidence-based and informed by scientifically rigorous research that has been conducted and replicated across the populations that would benefit the most from such interventions.

Despite the extant literature, gaps remain in the evidence for associations between key SSDOH (e.g., structural racism, policies, discrimination, neighborhood-level determinants) and cardiovascular disease and aging-related outcomes. Longitudinal cohort studies of middle to older age adults such as the Multi-Ethnic Study of Atherosclerosis (MESA) are well-poised to

address these gaps. Since 2000, MESA has been collecting rich data on SSDOH and cardiovascular and aging outcomes and biomarkers. Yet, many published MESA studies have not taken full advantage of these data to conduct more nuanced studies of SSDOH [12]. The aims of this paper are to: 1) describe the SSDOH measures and instruments available in MESA, 2) summarize MESA's contributions to the SSDOH field via a literature review of published papers as of January 23, 2023, and 3) identify gaps with respect to SSDOH measures and instruments and SSDOH-related publications that could be targeted for future MESA studies. Our literature review is based on the Schulz framework of Social Determinants of Health and Environmental Health Promotion, and thus we categorized the SSDOH according the categories outlined in the framework as follows: 1) macrosocial/structural factors (e.g., residential segregation and ideologies such as racism); 2) individual-level inequalities (e.g., employment, health insurance, income, and education); 3) built environment (e.g., land use, transportation access, and park space); 4) social context (e.g., neighborhood social cohesion and neighborhood socioeconomic status); 5) stressors (e.g., air pollution, chronic stress, violence and crime); 6) social integration and social support (e.g., social participation and social support) [13]. The overarching goals of the paper are to provide a guide on SSDOH for existing and new researchers using MESA data, and more broadly, to inform the larger research community on measures that could be added and SSDOH studies that could be conducted in other cohorts.

## Materials and methods

MESA is an ongoing, population-based, longitudinal prospective cohort study of subclinical cardiovascular disease (CVD) and progression to clinical disease among individuals enrolled at six US locations (Forsyth County, North Carolina; New York, New York; Baltimore, Maryland; Saint Paul, Minnesota; Chicago, Illinois; Los Angeles, California). Individuals were excluded if they had prevalent CVD at baseline. Details about MESA objectives and design are published elsewhere [12]. Briefly, the study was designed to be racially/ethnically diverse (28% Black, 12% Chinese, 22% Hispanic, and 39% White at Exam 1, based on self-report) and has conducted 7 in-person exams (Exam 1, 2000–2002; Exam 2, 2002–2004; Exam 3, 2004–2005; Exam 4, 2005–2007; Exam 5, 2010–2011; Exam 6, 2016–2018; Exam 7, 2022–2024) and 25 telephone follow-ups between the exams from 2000 to 2023. While CVD outcomes are of primary interest for MESA, numerous biomarkers and aging related outcomes (e.g., magnetic resonance imaging, inflammation markers, polysomnography, cognitive testing) have been collected during participant follow-up. The current study, as a literature review, used de-identified data from published papers and thus did not require Institutional Review Board review and did not require informed consent.

Our review began with an initial list of 2,125 papers identified by the MESA Coordinating Center that had been published using MESA data as of January 23, 2023. This list includes all papers (excluding a small portion of consortium papers) submitted to MESA's Publications and Presentations Committee for review and approval prior to journal submission (a requirement when using MESA data). The list was imported into Covidence systematic review software (Veritas Health Innovation) so that we could perform a systematic review of papers focused on SSDOH. Six papers were eliminated due to duplication, leaving 2,119 abstracts for initial review (Fig 1). Eight individuals reviewed the abstracts and papers moved from abstract screening to data extraction if the paper met any of the following criteria: 1) a SSDOH was a primary exposure variable; 2) a SSDOH was a primary outcome variable; 3) a primary aim (stated in the title, abstract, or introduction section) was to stratify analyses by an SSDOH or examine effect modification by an SSDOH (e.g., testing statistical interactions); or 4) the paper was descriptive or methodological in nature (i.e., not testing a hypothesis) but one or more

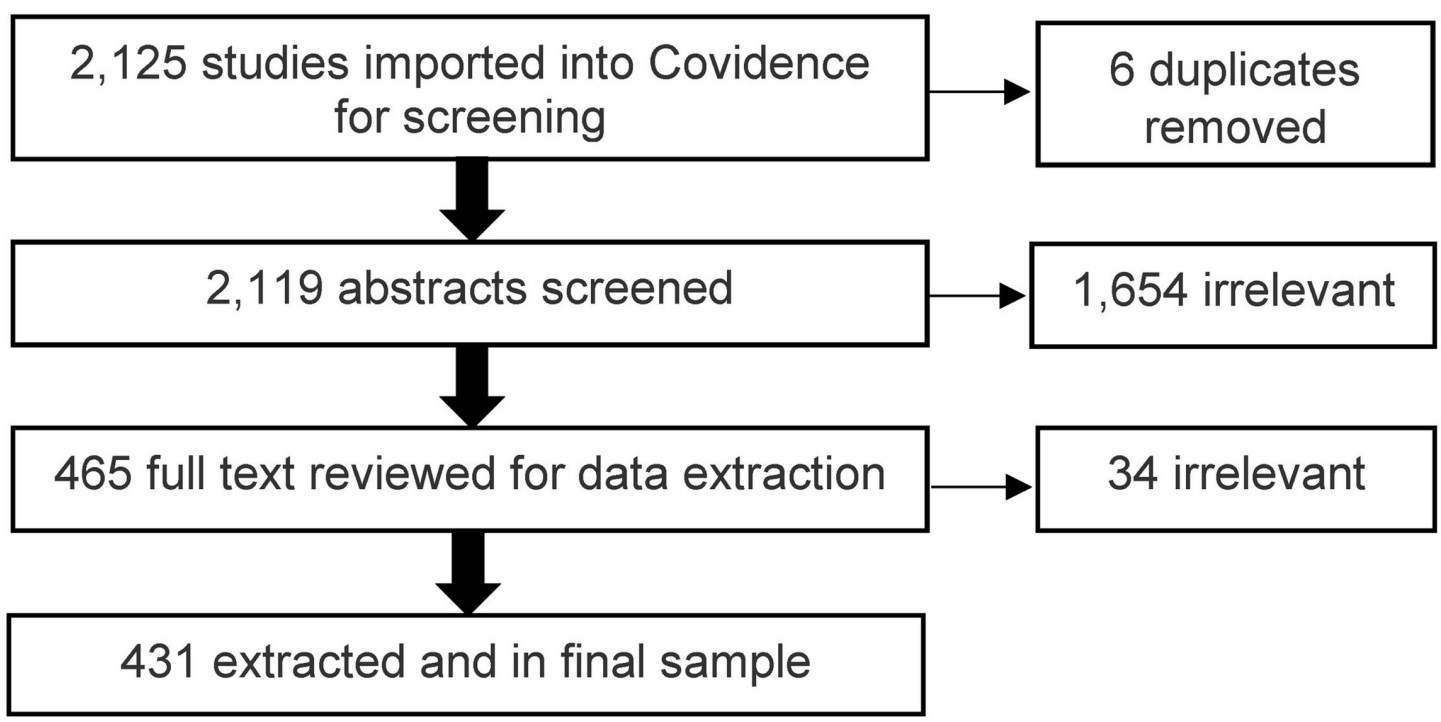

**Fig 1. Flowchart of literature review resulting in 431 studies using multi-ethnic study of atherosclerosis to study structural/social determinants of health.**

SSDOH were the main focus. Two reviewers were required for each paper and each step of the abstract and data extraction stages.

After abstract screening, 465 papers initially met eligibility criteria. Data were then extracted using a pre-determined template, which collected basic study data (title, author, year), whether the paper was focused only on methodology or only on air pollution modeling methods, whether the study stratified analyses by or examined effect modification by an SSDOH; whether MESA data were combined with other cohorts (e.g., MASALA); and the major categories of SSDOH included based on the major categories of determinants included in the Schulz framework (e.g., macrosocial/structural, individual-level inequalities, etc.) [13]. During the abstract screening and data extraction process, discrepancies between reviewers were resolved via a third-party tiebreaker/consensus by another reviewer from our team. The full-text review was conducted at the same time as the data extraction, such that papers (n = 34) were subsequently excluded during extraction if found to not meet our inclusion criteria listed above. Our review in Covidence resulted in a final sample of 431 papers with a SSDOH focus (20% of screened) (S1 File).

We then used the extracted data on the SSDOH categories to develop figures and tables outlining the number, timing, and categories of SSDOH measures used in MESA papers published as of January 23, 2023. Health outcomes were also extracted in 9 categories: (1) biomarkers (e.g., carotid ultrasound, magnetic resonance imaging), (2) cardiovascular disease (CVD) outcomes and risk factors (e.g., myocardial infarction, atrial fibrillation, hypertension), (3) lifestyle behaviors (e.g., diet and physical activity), (4) mental health (depression, anxiety), (5) cognition/dementia (e.g., process speed and global cognition test scores), (6) sleep outcomes (e.g., apnea, daytime sleepiness, actigraphy), (7) other health outcomes (e.g., mortality, obesity, cancer), (8) genetics (e.g., single nucleotide polymorphisms (SNP)), and (9) no health outcomes (e.g., air pollution methodology papers). Reported SSDOH and health outcome

categories are not mutually exclusive because some papers include multiple SSDOH and/or outcomes.

## Results

Table 1 provides the categories of SSDOH data collected at each of the seven completed MESA exams conducted from 2000–2023. SSDOH data were also collected at some telephone follow-up (FU) calls, which are conducted in between the in-person MESA exams and typically ask questions on general health, medical conditions and procedures, medications, admissions, and death. See the Supplemental Tables for more detailed descriptions of the SSDOH data collected, including SSDOH collected at FU calls.

### Macrosocial/structural measures

**Data captured in MESA.** Macrosocial/structural measures captured in MESA include sex/gender, racialized/ethnic group, language spoken, and acculturation measures (place of birth, years living in US), which were based on Exam 1 questionnaires (Tables 1 and S1). In addition, neighborhood measures of racial and ethnic segregation (i.e., Getis-Ord statistic) were calculated using geographic information systems (GIS) and are/will be available for addresses at Exams 1–7. We use the term "racialized" to indicate that race is socially constructed, and ethnic group refers to self-identified ethnicity (Hispanic or Chinese). Neither of these variables are biological.

**Macrosocial/structural papers.** Two hundred and forty-five papers (57% of total SSDOH sample) have been published using macrosocial/structural measures in MESA (Fig 2) [14–258]. Macrosocial/structural variables were most often (66%) stratification/effect modifier variables (versus exposures or outcomes) and racialized/ethnic group was the most common stratification variable (91%), followed by sex/gender (33%) (S2 Table). Both racialized/ethnic group and sex/gender were used together as stratifying variables in 23% of papers. Macrosocial/structural variables were exposures in 42% of papers (within those: racialized/ethnic group = 72%; acculturation = 21%; sex/gender = 20%, segregation = 7%). Macrosocial/structural factors were examined with biomarkers as the outcome 49% of the time, followed by CVD (26%), other health outcomes (18%), lifestyle behaviors (4%), cognition/dementia (4%), mental health (2%), sleep outcomes (3%), and respiratory outcomes (1%).

Papers using MESA data have contributed to the literature on acculturation, most often measured by language spoken at home, place of birth, and years lived in the U.S. These papers measured associations with subclinical CVD [67, 97], CVD risk factors [65, 75, 186], kidney function [59], hypertension [173], depression symptoms [49], cognitive test scores [83], and incident diabetes [188]. Overall, these studies have shown that greater acculturation is associated with worse health outcomes. Studies using data from MESA have also examined the hypothesized healthy immigrant effect by examining health risk factors such as BMI, waist circumference, and health behaviors and found mixed results that are dependent on factors such neighborhood environment, SES, and ethnic subgroup [19, 20, 215].

**Stratification by racialized/ethnic group.** The multi-ethnic nature of the MESA cohort allows for significant contributions to understanding various diseases and health processes both within and between racialized/ethnic groups. One hundred and forty-seven papers stratified by racialized/ethnic group. Of these, 45 (31%) examined racialized/ethnic differences in CVD outcomes, such as coronary heart disease, heart failure, atrial fibrillation, and stroke and CVD risk factors such as diabetes and hypertension. Racialized/ethnic group stratification has been used to examine the differential associations between residential segregation and neighborhood environment on cardiovascular outcomes such as incident CVD [139] and incident

**Table 1. Variables collected by structural and social determinants of health category.**

| SSDOH category | Variable/instrument collected (data source) | Exam | | | | | | |
|---|---|---|---|---|---|---|---|---|
| | | 1 | 2 | 3 | 4 | 5 | 6 | 7 |
| Macrosocial factors | Racialized / ethnic group (questionnaire) | X | | | | | | |
| | Sex/gender (questionnaire) | X | | | | | | |
| | Acculturation (e.g., place of birth) (questionnaire) | X | | | | | | X |
| | Neighborhood racial/ethnic segregation (GIS) | X | X | X' | X | X | * | * |
| Inequalities | Employment and health insurance (questionnaire) | X | X | X | X | X | X | X |
| | Income (questionnaire) | X | X | X | | X | X | X |
| | Investments, car, and property ownership (questionnaire) | | X | X | | X | | X |
| | Participant's education (questionnaire) | X | | | | | | X |
| | Mother and father's education (questionnaire) | | X | | | | | |
| | Own or rent residence (questionnaire) | X | X | X | | X | | X |
| Built environment | Physical activity/recreational environment (GIS or questionnaire) | X | X | X | X | X | * | * |
| | Food environment (GIS or questionnaire) | X | X | X | X | X | * | * |
| | Aesthetics (questionnaire) | | X | | | X | | X |
| | Population density/urbanicity (GIS) | X | X | X | X | X | * | * |
| | Street connectivity (GIS) | X | X | X | X | X | X | X |
| | Walking destinations/walkability (GIS) | X | X | X | X | X | * | * |
| | Transit access (GIS) | X | X | X | X | X | * | * |
| | Parks/greenspace (GIS) | X | X | X | X | X | X | * |
| Social context | Neighborhood social cohesion (questionnaire) | X | X | | | X | | X |
| | Changes to neighborhood social environment (e.g., demographics) (questionnaire) | | | | | | | X |
| | Neighborhood socioeconomic status (GIS) | X | X | X | X | X | * | * |
| | Neighborhood racialized/ethnic group composition (GIS) | X | X | X | X | X | * | * |
| | Neighborhood age composition/segregation (GIS) | X | X | X | X | X | * | * |
| Stressors | Neighborhood problems (safety, crime, violence, disorder, traffic, noise) (questionnaire) | X | X | | | X | | X |
| | Job demands (questionnaire) | X | X | X | | | X | X |
| | Emergency room medical care (questionnaire) | X | X | X | X | X | X | X |
| | Environmental tobacco smoke exposure (questionnaire) | X | | X | X | X | X | X |
| | Chronic burden/stress (questionnaire) | X | | X | X | | | |
| | Indoor or outdoor air pollutant exposure (e.g., use of wood to heat home, time spent on heavy traffic roads) (questionnaire) | | | | | X | X | X |
| | Discrimination (questionnaire) | X | | | | | | X |
| | Changes to neighborhood built and social environments suggestive of gentrification (questionnaire) | | | | | | | X |
| | Neighborhood crime (GIS) | X | X | X | X | X | | |
| | Outdoor air pollutants (e.g., PM$_{2.5}$, O$_3$) (GIS) | X | X | X | X | X | * | * |
| Social support/ social integration | Social participation (questionnaire) | | X | X | | | | X |
| | Household characteristics (e.g., number, marital status) | X | | X | X | X | | X |
| | Volunteer work/caring for others | X | X | X | | X | X | X |
| | Social support (e.g., someone to count on) | X | | X | X | X | | |
| | Social isolation | | | | | X | X | |

* Planned but not yet calculated

GIS = Geographic Information System (i.e., objective measures); SSDOH = Structural and social determinants of health

Note: Major categories reported in this table, with additional categories and more specific measures reported in the supplemental tables. Exam calendar years: 1, 2000–2002; 2, 2002–2004; 3, 2004–2005; 4, 2005–2007; 5, 2010–2011; 6, 2016–2018; 7, 2022–2024.

hypertension [100], health behaviors such as physical activity, diet quality, and alcohol consumption [41, 42, 68, 99], cardiometabolic risk [163] as well as on cognitive outcomes [33, 259, 260]. For example, Gao et al, 2022 [99] found that a composite measure of healthy

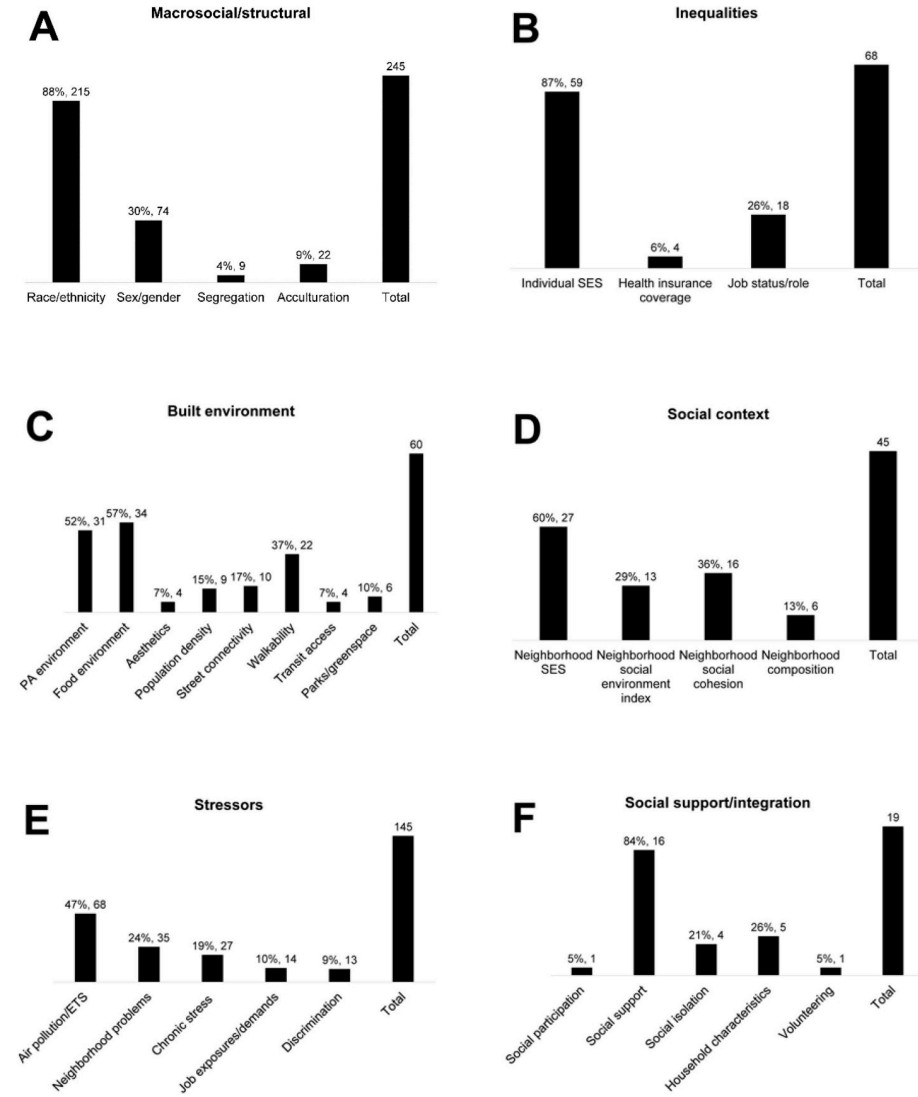

**Fig 2. Papers in each structural and social determinants of health.** Abbreviations: SES = socioeconomic status; PA = physical activity; ETS = environmental tobacco smoke.

neighborhood food environment was associated with better diet quality for all participants but that the association was twice as high within participants racialized as Black as it was within participants racialized as White. Three recent studies in MESA have shown that residential segregation is associated with worse health including cognitive decline [34], hypertension [100], and cardiometabolic risk [163]. Using stratification, two reports found that the associations were more pronounced among participants racialized as Black and Hispanic and one found this among only participants racialized as Black. Another study found that living in areas that were historically redlined was associated with worse cardiovascular health, and that the association was more pronounced among participants racialized as Black [175].

Research using MESA has also contributed to literature attempting to disentangle racialized/ethnic group effects and socioeconomic effects, a longstanding issue in health disparities research. One study showed that the expected racialized/ethnic differences were not found but differences by income were pronounced [42]. Another study found inverse socioeconomic

patterning of outcomes such as diabetes and hypertension among women racialized as Black and White but mixed relationships among men as well as differing associations depending on the definition of SES (income vs. education) [40].

### Individual-level inequality measures

**Data captured in MESA.** Inequality measures include numerous measures of individual-level socioeconomic status (e.g., income, education of participant and parents, occupation, home, property, and car ownership) (Tables 1 and S3). Health insurance type, number of people supported by household income, and owning/renting a home are also collected. In addition, at telephone follow-up 23, MESA participants are asked a series of questions on early life educational quality (e.g., single room schoolhouse, segregated school) (S3 Table) [261, 262].

**Inequality papers.** A total of 68 papers (16% of total) focused on individual-level inequalities in three domains: individual socioeconomic status (87%), health care coverage (6%), and employment status/occupational role (26%) (Fig 2 and S4 Table) [5, 23, 35, 39–41, 47, 50, 51, 53, 82, 89, 92, 95, 108, 111, 116, 127, 156, 179, 195, 202, 222, 233, 239, 253, 263–304]. These were modeled as the primary exposure in 74% of the inequalities focused papers and as a stratifying factor in 29% of papers, while no papers included it as an outcome. The three inequality domains were most frequently assessed in relation to biomarkers (43%), lifestyle factors (18%) and other health outcomes (24%) (Table 2). No papers examined associations between inequality measures and sleep outcomes.

Greater individual and parental educational attainment and household income were frequently associated with better clinical, biological, and environmental health measures, positive health behaviors, and lower rates of disease for MESA participants. Consistently, access to health insurance was associated with better health outcomes [82, 195, 278, 284]. Generally, greater wealth was associated with better health outcomes, though a few studies did observe

**Table 2. Health outcomes examined in MESA studies of structural and social determinants of health.**

| Health outcome | Number (%) of studies by SSDOH category | | | | | | |
|---|---|---|---|---|---|---|---|
| | Row total (col %) | Macrosocial/structural (col %) | Inequalities (col %) | Built environment (col %) | Social context (col %) | Stressors (col %) | Social support/ integration (col %) |
| Biomarkers | 204 (47%) | 119 (49%) | 29 (43%) | 3 (5%) | 12 (27%) | 75 (52%) | 8 (42%) |
| CVD outcomes/risk factors | 82 (19%) | 63 (26%) | 6 (9%) | 9 (15%) | 6 (13%) | 16 (11%) | 4 (21%) |
| Lifestyle behaviors | 38 (9%) | 12 (4%) | 12 (18%) | 22 (37%) | 6 (13%) | 8 (6%) | 2 (11%) |
| Respiratory outcomes | 6 (1%) | 3 (1%) | 1 (1%) | 0 (0%) | 0 (0%) | 3 (2%) | 0 (0%) |
| Mental health | 12 (3%) | 6 (2%) | 3 (4%) | 5 (8%) | 7 (16%) | 5 (3%) | 1 (5%) |
| Cognition/dementia | 13 (3%) | 9 (4%) | 4 (6%) | 5 (8%) | 0 (0%) | 1 (1%) | 1 (5%) |
| Sleep outcomes | 13 (3%) | 8 (3%) | 0 (0%) | 3 (5%) | 2 (4%) | 3 (2%) | 0 (0%) |
| Other health outcomes | 71 (16%) | 44 (18%) | 16 (24%) | 14 (23%) | 14 (31%) | 13 (9%) | 4 (21%) |
| Genetics | 9 (2%) | 1 (0%) | 4 (6%) | 0 (0%) | 1 (2%) | 5 (3%) | 2 (11%) |
| No health outcomes* | 31 (7%) | 6 (2%) | 5 (7%) | 5 (8%) | 3 (7%) | 25 (17%) | 0 (0%) |
| Total papers (row %) | 431 (100%) | 245 (57%) | 68 (16%) | 60 (14%) | 45 (10%) | 145 (34%) | 19 (4%) |

CVD = cardiovascular disease; SSDOH = Structural and social determinants of health

Note: SSDOH and health outcome categories are not mutually exclusive

* e.g., focused on methods

unexpected worse health outcomes with indicators of greater wealth [291, 299]. The transition to retirement was associated with decreases in overall moderate/vigorous physical activity, but increases in leisure walking, all of which may differ by accrued income [127, 288].

## Built environment

**Data captured in MESA.** As part of the grant-funded MESA Neighborhood ancillary studies, built environment measures in MESA were captured through three primary channels: (1) derived through Geographic Information Systems (GIS); (2) questionnaires to participants; and (3) a series of companion questionnaires to residents (i.e. community informants) living in the same neighborhoods as participants but not part of MESA (known as the "Community Survey") which were then processed to create Bayesian estimates for MESA neighborhoods (Tables 1 and S5). The GIS measures broadly capture land uses (e.g., retail, residential and park space), density (e.g. population density, urbanicity), connectivity for travel (e.g. street connectivity, transit access), and destinations (e.g. walkability, destinations for social engagement, resources for recreation, food stores). GIS data exist across varied neighborhood definitions and scales including census tract and both Euclidean/network buffers of many distances. Data collected via questionnaires (both MESA and Community Survey) include access to places and amenities in the neighborhood (e.g., grocery stores, exercise facilities, parks, places to learn, places for older adults, public art), presence of neighborhood walking supports (e.g. sidewalks and pedestrian amenities such as crosswalks), neighborhood aesthetics, residence (e.g., single family, assisted living) and building type (e.g., manufactured home, condo), and indoor home facilities (e.g., air conditioning, window locations, stove/range type). Several validated scales have been constructed from these questionnaires with the most prominent being walkability, healthy food access, and aesthetics as well as three outside of built environment (i.e. safety, disorder, and social cohesion). Many built environment measures through Exam 5 are available for work addresses in addition to primary residential addresses. Additionally, during telephone follow-up call 14, MESA participants were asked their residential history focused on primary addresses.

**Built environment papers.** Sixty papers (14% of total SSDOH papers) included at least one measure of the built environment (Fig 2) [20, 33, 35, 41, 42, 68, 99, 100, 111, 116, 123, 127, 158, 175, 187, 227, 259, 260, 271, 305–345]. A majority of papers modeled built environment as exposures (88%) with only 5% examining them as outcomes and 10% using them for effect measure modification (S6 Table). Food environment and physical activity or recreation environment (either GIS or survey-based measures) were the most common built environment metrics, appearing in 57% and 52% of papers, respectively. Walking destinations and walkability (via GIS) were similarly well represented, with 37% of papers examining these measures. While less common, many studies used street connectivity (17%), density/urbanicity (15%), parks/greenspace (10%), aesthetics (7%), and transit access (7%). No papers focused on health care access (e.g., density of health care facilities in the neighborhood). The built environment domain measures were most frequently assessed in relation to lifestyle factors such as physical activity or nutrition (37%) and cardiovascular disease outcomes or risk (15%) (Table 2). However, MESA built environment work covers a variety of outcomes including mental health (8%), cognition/dementia (8%), sleep (5%), or biomarkers (5%). Although some papers combined built environment with genetic data, no papers examined built environment with genetic outcomes (e.g., DNA methylation). Similarly, no work examined built environment with respiratory outcomes.

The measurable impact of MESA built environment data on the field of neighborhoods and disease have been discussed elsewhere [346]. However, briefly, MESA has found robust

evidence of the positive relationship between complementary measures of healthy food environments (individual perceptions, neighborhood informants, store densities) and a healthy diet (e.g., [271, 334, 335, 347, 348]). Similarly, MESA has provided strong evidence that participants who live in supportive neighborhood environments (higher walkability, more recreation destinations, higher density) report higher levels of walking and other physical activity (e.g., [276, 322, 341, 349]). Residents of neighborhoods with worse walking environments had higher odds of severe sleep apnea (CIT), although neighborhood characteristics were not associated with insomnia, sleep efficiency, or sleep fragmentation [314, 316, 350, 351]. Lower social engagement destinations and poorer aesthetic quality have been associated with depressive symptoms and incident depression [330, 333]. Related to lifestyle factors, built environment measures in MESA have been associated with lower hypertension, lower body mass index, and lower insulin resistance and impaired fasting glucose [323, 352, 353]. Evidence on built environments and cognitive function have been mixed: additional greenness has been associated with better cognitive outcomes while increasing density (destination density, street density) were associated with worse overall cognition (e.g., [35, 260, 354, 355]). Similarly, evidence between built environments and cognitive function have been mixed depending on outcome (i.e. global cognitive function, processing speed). Clarifying these associations and competing pathways is the emphasis of the most recent MESA Neighborhood ancillary study [111].

## Social context measures

**Data captured in MESA.** Social context measures collected via questionnaire include neighborhood friendliness, neighborhood social cohesion (e.g., close knit neighborhood, trust in neighbors), and changes to the neighborhood social environment (i.e., Perceptions About Changes in Environments and Residents questionnaire (PACER [356])) (Tables 1 and S7). GIS-based measures derived for the MESA Neighborhoods ancillary studies include neighborhood SES, neighborhood racialized/ethnic group, and age group composition. In addition, informants living in the neighborhoods where MESA participants lived completed surveys at Exams 3, 4, and 5 on various aspects of the neighborhood social environment (e.g., neighbors willing to help one another).

**Social context papers.** Forty-five papers (10% of total SSDOH papers) have been published using MESA's social context measures (Fig 2) [5, 41, 60, 70, 74, 111, 116, 175, 187, 265, 266, 269, 286, 289, 295–298, 302, 303, 316, 327, 330, 332, 333, 336, 344, 351, 357–373]. Of those, 89% investigated social context as the exposure variable (of those: 63% on NSES, 25% on neighborhood social environment, 38% on neighborhood social cohesion, and 15% on neighborhood composition) (S8 Table). Fewer studies examined social context as an outcome (7%) or as a stratifying variable/effect modifier (11%). Papers investigated associations between social context and biomarkers (27%), CVD outcomes or risk factors (13%), lifestyle factors (13%), mental health (16%), and sleep outcomes (4%), while no studies examined associations with respiratory or cognitive/dementia outcomes.

Recently published papers evidenced the significant contribution of MESA to not only to the field of SSDOH research, but also in going beyond traditional cardiovascular/cerebrovascular health outcomes to link SSDOH to aging biomarkers. For example, these include studies of the socioeconomic gradient in epigenetic aging clocks; social support, psychosocial risks, and cardiovascular health; and discrimination, neighborhood social cohesion, and telomere length [296, 298, 374]. In the first study, in which MESA data were combined with the Health and Retirement Study (HRS), the authors found that individuals living in neighborhoods with higher socioeconomic disadvantage demonstrated accelerated biological aging measured via the GrimAge and DunedinPoAm epigenetic clocks [298]. Social support was not associated

with 7 CVH indicators in the second study, which combined data from MESA with the Jackson Heart Study and the Mediators of Atherosclerosis in South Asians Living in America Study (MASALA) [296]. In the third study, neighborhood social cohesion modified the association between discrimination and telomere length attrition over time (i.e., those in low cohesion neighborhoods experiencing more discrimination had faster telomere attrition) [374].

## Stressor measures

**Data captured in MESA.** Stressor measures collected by questionnaire include neighborhood safety (e.g., violence, crime); neighborhood psychosocial disorder (e.g., trash, noise); job demands, job control, and job security; environmental tobacco smoke at home, work, and during childhood; obtaining medical care at emergency rooms; chronic stress burden (e.g., health problem, job problem, financial strain); traffic exposure; lifetime and everyday discrimination; noise inside and outside of the home that interrupts sleep; and changes to the neighborhood built and social environment (e.g., demographic shifts suggesting gentrification) (Tables 1 and S9). During telephone follow-ups 9, 10, and 13, MESA participants answered a questionnaire about residence type (e.g., single family, manufactured mobile home) and in-home and travel behaviors that would affect indoor and outdoor air pollution exposure (e.g., air filter and air conditioning use, locations out of the home spent ≥2 hours/day). Community informants living in MESA participants' neighborhoods completed surveys at Exams 3, 4, and 5 on stressors such as feeling safe walking in the neighborhood day or night and a lot of graffiti in the neighborhood. In addition, exposure to early life adversity (ELA) was examined at telephone follow-up 20. Items were classified as a measure of ELA under the following domains: parental support/affect; emotional abuse; parental physical affection; physical abuse; household substance abuse; household organization; and parental monitoring.

**Stressor papers.** One hundred and forty-five papers (34% of total SSDOH papers) have been published using MESA's stressor measures (Fig 2) [5, 25, 30, 39, 50–53, 57, 66, 74, 79, 87, 92–95, 110, 111, 116, 124, 125, 149, 164, 174, 193, 223, 226, 227, 255, 263, 267, 279, 280, 282, 286, 299, 300, 315, 316, 319, 326, 327, 330, 333, 336, 338, 343, 344, 351, 357, 367, 372, 375–466]. Of those, 47% included air pollution and/or environmental traffic measures, 24% included measures of neighborhood stressors (problems, safety, crime, disorder), 19% included measures of chronic stress, 10% included measures of occupational exposures or job demand, and 9% included measures of discrimination (S10 Table). Most papers (89%) included the stressor variables as the exposure or predictor and far fewer as the outcome (9%) or as a stratification/effect modification variable (8%) Over half (52%) of the papers studied stressor measures in relation to biomarker outcomes, with smaller percentages dedicated to CVD-related outcomes (11%), other health outcomes (9%), lifestyle behaviors (6%), or the other SSDOH categories (i.e., 2% on respiratory outcomes, 3% mental health, 1% cognition/dementia, 2% sleep, 3% genetics, 17% did not include health outcomes (e.g., air pollution methodology)) (Table 2).

The greatest percentage of the papers on stressors (47%) were on air pollution or environmental tobacco smoke (ETS). Twenty-five papers (17% of total focused on stressors) did not study air pollution or other SSDOH as predictors, outcomes, or stratifying/effect modifier variables, but solely reported on air pollution estimate methodology. Importantly, the EPA-funded MESA Air ancillary study led to new spatiotemporal techniques for improved estimates of outdoor air pollutants (e.g., $PM_{2.5}$, $O_3$) [429, 467]. These models aimed at capturing pollutant variation in metropolitan areas with a focus on small temporal and spatial scales, and over time the methodology led to the ability to estimate fine-scale exposures across the US, which in turn can be studied in relation to numerous health outcomes.

Recent MESA studies interested in associations with air pollution have shown that short term exposure to air pollution is associated with increased respiratory infections [428], increased biomarkers for vascular disease [442], increased risk of emphysema [464], and increased likelihood of sleep apnea [390]. One study found that exposure to air pollution accounted for some of the racialized/ethnic group disparities in systolic blood pressure among men [226] while another found no association between air pollution and blood pressure when accounting age and calendar time [376]. Overall, MESA studies mostly show that exposure to short- and long-term air pollution is detrimental to health. Recent MESA papers of ETS exposure found that secondhand smoke exposure is associated with higher risk of atrial fibrillation in non-smokers [433] and that ETS in childhood is associated with more emphysema in adulthood [436]. One study found that working in blue collar workplaces was associated with heavy smoking and with ETS.

Papers that included chronic stress (measured by the chronic burden scale) as a predictor for health-related outcomes reveal mixed findings. Chronic stress was positively associated with incident ASCVD and mortality [382], incident stroke and TIA [404], CVD events [397], endothelial dysfunction [424], central and subcutaneous adiposity [400], and inflammation [450]. Other papers found negative associations between chronic stress and inflammation [223], urinary stress hormones [50], coronary artery calcification [401], and atrial fibrillation [405]. Associations between chronic stress and health-related outcomes may depend on type and length of exposure to stress.

## Social support/social integration

**Data captured in MESA.** Social support and social integration measures collected via questionnaire include marital status; number in household; living situation (alone, with others); time spent caring for a child/adult; volunteer work; frequency attending religious services; participation in neighborhood organizations (e.g., block association, sport leagues); social support (e.g., someone to give advice, someone to provide love and affection); social isolation; and provided or received emotional/unpaid assistance to or from others (Tables 1 and S11).

**Social support papers.** Nineteen papers (4% of total SSDOH papers) have been published using MESA's social support and social integration measures (Fig 2) [83, 116, 127, 185, 267, 268, 296, 361, 407, 416, 440, 468–475]. Eighty-four percent of publications used these measures as the primary exposure and 37% used social support/social integration measures as stratifying/effect modifier variables (S12 Table). Primary exposures included social support (84%), social isolation/loneliness (21%), household size or marital status (26%), caregiving (5%), and social participation (5%). Among papers that stratified by social support/integration variables (n = 7), 71% stratified by social support measures and 43% of stratified by household characteristics (e.g., marital status). No other measures were used for stratification/effect modification analyses.

Notably, MESA social support and social integration measures were most frequently examined for associations with mechanistic factors, specifically biomarkers (42%), followed by CVD outcomes/risk factors (21%). With respect to biomarkers, Brown et al. (2020) found that loneliness was associated with inflammatory biomarker gene expression [268]. Low social support was associated with C-reactive protein (CRP) elevation in men but not in women, though high social support buffered the relationship between stress and CRP among middle-aged women [473]. In addition, a positive association was found between social support and telomere length [473]. Four publications examined social support and social integration measures in the context of CVD risk or CVD outcomes, with only one publication finding a positive association between social support and time delay in CVD disease onset [468].

Eleven percent of publications examined the association of social support with other health outcomes. Lower social support was associated with higher rates of retinopathy, social support was not associated with metabolic syndrome prevalence, and social support was inversely associated with heart rate but not heart rate variability [185, 416, 475]. Another study found a significant association between household size and cognitive test scores, with those living in households with 5 members or more scoring lower on cognitive testing [83]. Lastly, no association was found between social support, change in caregiver status, or marital status, and walking post-retirement [127]. Overall, few MESA studies have examined social support/integration measures in relation to health-related outcomes.

### Additional MESA measures associated with SSDOH

MESA has also collected data that help provide further details regarding the aforementioned SSDOH measures or that allow for the derivation of new neighborhood GIS measures (S13 Table). This includes years lived in the neighborhood, age/birth date (can determine birth cohort/cohort effects); family members' sex/gender including spouse/partner; residential history back to 1980 and secondary residence address history (e.g., snowbirds); and time spent doing activities in the neighborhood. The residential history data allows interested investigators to create new neighborhood measures by linking geocoded addresses (latitudes and longitudes) to environmental data (e.g., satellite imagery, google earth imagery, maps).

### Combination of MESA with other cohorts

Forty-seven papers (11% of total SSDOH papers) combined MESA with 41 US and non-US based cohorts [5, 31, 44, 61, 80, 85, 88, 90, 102, 107, 112, 130–132, 134, 135, 152, 159, 168, 180, 191, 216, 220, 245, 256, 293, 295, 296, 298, 300, 304, 399, 476–490] (S14 Table). Among these 47 papers, 85% included macrosocial measures, 15% included inequality measures, 0% included built environment measures, 11% included social context measures, 15% included stressor measures, and 6% included social support measures (percentages not mutually exclusive since some papers include more than one SSDOH). In comparison to the full sample of papers, those combining MESA data with other cohorts more often included macrosocial measures (85% vs 57%), fewer (none) included built environment measures (0% versus 14%), and fewer included stressor measures (15% versus 34%) (Table 2).

The most common cohorts combined with MESA include the Mediators of Atherosclerosis in South Asians Living in America Study (MASALA), Atherosclerosis Risk in Communities Study (ARIC), Jackson Heart Study (JHS), and the Health and Retirement Study (HRS) (S14 Table). For instance, in a recently published paper (2022), MESA data were combined with 14 other cohorts in Africa, Asia, Europe, and North America to investigate whether associations between common cardiovascular risk factors and common carotid intima-media thickness (CIMT) varied by racialized/ethnic group [180]. The study found that high density lipoprotein levels (HDL) were protective only for African American and African individuals. For Asian populations, smoking and glucose displayed the strongest associations with CIMT. The study noted variations in the strength of associations between the risk factors and CIMT depending on racialized/ethnic group, suggesting that primary prevention strategies need to be tailored to each group. Overall, MESA's comprehensive data collected on a diverse sample allows for these pooled and comparison studies that have contributed significantly to the literature.

### Discussion

MESA has contributed over 430 SSDOH papers to the extant literature since its inception. Consistent with MESA's aim of assembling and following a diverse, multi-ethnic cohort that

allows for analyses of specific racialized/ethnic subgroups, approximately half of the studies involved racialized/ethnic group as an exposure or stratifying/effect modifying variable. Moreover, hundreds of published MESA papers focused on a wide range of SSDOH, investigating macrosocial/structural factors such as acculturation and neighborhood racialized/ethnic segregation, individual-level inequalities such as health insurance coverage and retirement status, built environment factors such as greenspace and destinations, social context factors such as neighborhood SES and social cohesion, stressors such as discrimination and air pollution, and social support/integration factors such has loneliness and social participation. MESA's SSDOH data has been combined with 41 other cohorts across 47 papers for pooled analyses, cross-cohort comparisons, and replication or validation of instruments and algorithms/indices. Over time, additional self-reported SSDOH instruments (e.g., PACER, age-related neighborhood characteristics) questionnaires have been incorporated into MESA exams. These additions are evidence of MESA's acknowledged importance of understanding how SSDOH impact CVD and aging-related outcomes.

## Gaps in SSDOH related papers to target in future research

The least studied SSDOH category was social support/integration, suggesting avenues for future research. In particular, only four papers focused on social isolation and loneliness, which was named as a US epidemic and as risk factors for numerous poor health outcomes, including mortality, cardiovascular disease, and cognitive decline in the 2023 US Surgeon General's report [491]. New research could investigate MESA's social support/integration measures such as isolation and loneliness as risk factors for CVD and aging outcomes and as mediators or outcomes in relation to other SSDOH risk factors such as retirement status and urbanicity.

Sleep behaviors and patterns can be altered in older age [492]. Older adults can experience sleep apnea, increased daytime sleepiness, and more frequent naps, as well as shorter length of time slept or disrupted sleep at night. While MESA has collected several self-reported (e.g., snoring, nap frequency) and objective measures of sleep (e.g., actigraphy), only 13 of the reviewed studied examined associations between SSDOH and sleep outcomes. Six of these focused on racialized/ethnic group or other demographic comparisons of sleep outcomes, and four examined associations with neighborhood built and social environments. Future MESA studies could center on understudied SSDOH such as noise and social engagement and whether these impact sleep outcomes, in addition to whether the sleep measures mediate or moderate associations between risk factors (e.g., physical activity) and CVD and aging outcomes.

Respiratory outcomes such as COPD and asthma were the least studied health outcomes among MESA's SSDOH studies (n = 6). These papers studied outdoor air pollution, smoking, and SES as predictors of respiratory outcomes, occupational risk for COPD, and lung function prediction equations (i.e., spirometry). None examined indoor air pollution, possible structural (e.g., segregation) or neighborhood determinants of respiratory disease (e.g., neighborhood SES, neighborhood greenspace), which would be fruitful targets for future research.

Up to 40% of dementias including Alzheimer's disease have been attributed to modifiable risk factors (e.g., physical activity, diet, obesity, diabetes, depression, smoking, and hearing loss) that are heavily influenced by SSDOH [3]. To date, only 13 of MESA's SSDOH studies centered on cognitive/dementia outcomes. With its numerous SSDOH measures, MESA is well poised to investigate nuanced relationships between SSDOH, modifiable risk factors, and cognitive and dementia outcomes, to better elucidate the contribution of SSDOH to dementia risk.

The importance of intersectionality has been increasingly emphasized in SSDOH and health research. Individuals who have multiple, historically disadvantaged and minoritized identities (e.g., Black women or Hispanic individuals without a high school education) can be at increased risk for numerous diseases and health conditions including CVD and dementia [493, 494]. Yet, no MESA studies as of January 2023 have investigated and framed their studies based on the concept of intersectionality. This is an important next step for SSDOH and health research generally and future MESA studies specifically.

Structural determinants of health include laws, economic and social policies, and governing practices at local, regional, and national scales that determine the distribution of economic and health promoting resources and opportunities. To date, only two MESA studies examined social/economic policies in relation to health outcomes. The first investigated neighborhood cigarette prices and smoking ban policies on smoking behavior, finding that a $1 increase in price was associated with a 20% increased risk of smoking cessation [162]. MESA's residential address history data provides numerous opportunities for future studies to determine associations between policies at the participants' neighborhood, county, and state level (e.g., historic redlining) and CVD and aging outcomes. The second paper studied historic redlining (discriminatory mortgage lending practices) and cardiovascular health, finding that Black individuals living in historically redlined areas had worse cardiovascular risk scores based on blood pressure, fasting glucose, cholesterol, body mass index, diet, physical activity, and smoking [175].

Recent SSDOH additions in MESA will become available to researchers in the coming years. This includes self-reported data from the PACER, ELA, and early-life educational quality questionnaires, as well as objectively measured (i.e., GIS-derived) data on google street view (GSV) greenery (i.e., grasses, trees, shrubs) calculated for 2007-on (NIA 5R00AG066949, PI: Pescador), and measures of neighborhood greenness and greenspace (i.e., normalized difference vegetation index and developed open space, forest, and total greenspace derived from the National Land Cover Dataset) that were calculated annually for participants' addresses from 1980–2019 (NIA R21AG075291, PI: Besser). In addition, the ongoing ancillary MESA Neighborhoods III study (NIA R01AG072634, PI: Hirsch) is developing new survey measures on whether MESA respondents have neighborhood supports for aging as well GIS measures such as age segregation and American Association of Retired Persons (AARP) Livability Index domains (e.g., affordable housing, transportation safety and convenience, clean air and water) [356, 495]. The newly added measures will offer researchers numerous opportunities to collaborate with MESA investigators and investigate novel research hypotheses related to community-level measures of social and built environments.

## Strengths and limitations

Beyond the contributions and strengths noted above, MESA displays several strengths including its population-based recruitment, the multi-ethnic and geographically diverse sample, and the comprehensive data collected at each exam on self-reported and objectively measured SSDOH as well as on clinical and biomarkers outcomes. While the strengths of MESA's SSDOH data far outweigh any weaknesses, some common limitations must be noted. The administration timing for some SSDOH instruments (e.g., PACER) limits the analytic sample size because they were added during later exams. As with any longitudinal cohort, MESA has experienced attrition due to death and drop out. For instance, at Exam 7 (2022–2023), 34% (n = 2,317) of the original cohort of 6,814 from Exam 1 (2000–2002) remained. Focusing on SSDOH collected at Exam 7 may result in biased findings if analyses using these data do not account for selection/attrition bias, particularly for differential follow-up by SSDOH factors. Any self-reported SSDOH measures (e.g., questions on psychosocial disorder) may be subject

to misclassification or recall bias. Although the sample is geographically and racially/ethnically diverse, any associations found using MESA may not be generalizable to other racialized/ethnic groups or other regions in the US. MESA originally collected data on gender (i.e., male and female), but data are not collected on third/non-binary gender, biological sex assigned at birth, or sexual orientation. Data on secondary/additional racialized/ethnic identities are also not collected (e.g., self-identifying as Black and White). Lastly, the strong correlation between some SSDOH variables (e.g., racial/ethnic segregation and neighborhood SES) should be carefully weighed to determine if statistically controlling for the highly correlated variable or examining effect modification by that variable is feasible.

In this review, we evaluated the major categories of SSDOH collected and investigated in MESA to date, identified the scientific gaps, and outlined numerous possibilities for researchers to pursue in future SSDOH-related studies. New MESA studies can investigate previously unexplored areas by incorporating measures of social support/integration; centering on sleep, respiratory, and cognitive/dementia outcomes; examining structural determinants as well as SSDOH as mediators and moderators; and studying health outcomes with an intersectionality lens. Overall, MESA has substantially added to the extant literature on SSDOH and CVD and aging-related outcomes and provides a valuable resource for researchers interested in pursuing studies on SSDOH and older adults.

## Supporting information

**S1 Table. Macrosocial/structural measures collected by MESA exam.**
(DOCX)

**S2 Table. Papers with a focus on macrosocial/structural variables.**
(DOCX)

**S3 Table. Inequality measures collected by MESA exam.**
(DOCX)

**S4 Table. Papers with a focus on inequalities.**
(DOCX)

**S5 Table. Built environment measures collected by MESA exam.**
(DOCX)

**S6 Table. Papers with a focus on built environment variables.**
(DOCX)

**S7 Table. Social context measures collected by MESA exam.**
(DOCX)

**S8 Table. Papers with focus on social context variables.**
(DOCX)

**S9 Table. Stressor measures collected by MESA exam.**
(DOCX)

**S10 Table. Papers with focus on stressors.**
(DOCX)

**S11 Table. Social integration/support measures collected by MESA exam.**
(DOCX)

**S12 Table. Papers with a focus on social support/social integration.**
(DOCX)

**S13 Table. Other social determinants of health-related measures collected by MESA exam.**
(DOCX)

**S14 Table. Papers that combined MESA with other cohorts.**
(DOCX)

**S1 File. Details on SSDOH data extracted from the 431 papers.**
(XLSX)

# Acknowledgments

The MESA cohort study was supported by contracts 75N92020D00001, HHSN268201500003I, N01-HC-95159, 75N92020D00005, N01-HC-95160, 75N92020D00002, N01-HC-95161, 75N92020D00003, N01-HC-95162, 75N92020D00006, N01-HC-95163, 75N92020D00004, N01-HC-95164, 75N92020D00007, N01-HC-95165, N01-HC-95166, N01-HC-95167, N01-HC-95168 and N01-HC-95169 from the National Heart, Lung, and Blood Institute, and by grants UL1-TR-000040, UL1-TR-001079, and UL1-TR-001420 from the National Center for Advancing Translational Sciences (NCATS). The authors thank the other investigators, the staff, and the participants of the MESA study for their valuable contributions. A full list of participating MESA investigators and institutions can be found at http://www.mesa-nhlbi.org. This paper has been reviewed and approved by the MESA Publications and Presentations Committee.

# Author Contributions

**Conceptualization:** Lilah M. Besser, Sarah N. Forrester, Jana A. Hirsch.

**Data curation:** Lilah M. Besser.

**Formal analysis:** Lilah M. Besser, Sarah N. Forrester, Milla Arabadjian, Michael P. Bancks, Margaret Culkin, Kathleen M. Hayden, Isabelle Pierre-Louis, Jana A. Hirsch.

**Methodology:** Lilah M. Besser, Sarah N. Forrester, Jana A. Hirsch.

**Project administration:** Lilah M. Besser.

**Supervision:** Lilah M. Besser.

**Writing – original draft:** Lilah M. Besser, Sarah N. Forrester, Milla Arabadjian, Michael P. Bancks, Margaret Culkin, Elaine T. Le, Jana A. Hirsch.

**Writing – review & editing:** Lilah M. Besser, Sarah N. Forrester, Milla Arabadjian, Michael P. Bancks, Jana A. Hirsch.

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
