## [Decision Letter · Decision Letter 0]

11 Sep 2024

PONE-D-24-30988Structural and Social Determinants of Health: The Multi-Ethnic Study of AtherosclerosisPLOS ONE

Dear Dr. Besser,

Thank you for submitting your manuscript to PLOS ONE. After careful consideration, we feel that it has merit but does not fully meet PLOS ONE’s publication criteria as it currently stands. Therefore, we invite you to submit a revised version of the manuscript that addresses the points raised during the review process.

We look forward to receiving your revised manuscript.

Kind regards,

Andrea Da Porto

Academic Editor

PLOS ONE

Journal Requirements:

1. When submitting your revision, we need you to address these additional requirements. Please ensure that your manuscript meets PLOS ONE's style requirements, including those for file naming. The PLOS ONE style templates can be found at  https://journals.plos.org/plosone/s/file?id=wjVg/PLOSOne_formatting_sample_main_body.pdf and https://journals.plos.org/plosone/s/file?id=ba62/PLOSOne_formatting_sample_title_authors_affiliations.pdf.

Reviewers' comments:

Reviewer's Responses to Questions

**Comments to the Author**

1. Is the manuscript technically sound, and do the data support the conclusions?

Reviewer #1: Yes

Reviewer #2: Yes

Reviewer #3: Partly

Reviewer #4: Yes

2. Has the statistical analysis been performed appropriately and rigorously? 

Reviewer #1: Yes

Reviewer #2: Yes

Reviewer #3: N/A

Reviewer #4: Yes

3. Have the authors made all data underlying the findings in their manuscript fully available?

Reviewer #1: Yes

Reviewer #2: Yes

Reviewer #3: Yes

Reviewer #4: Yes

4. Is the manuscript presented in an intelligible fashion and written in standard English?

Reviewer #1: Yes

Reviewer #2: Yes

Reviewer #3: Yes

Reviewer #4: Yes

5. Review Comments to the Author

Reviewer #1: 1.The manuscript is technically sound, and the data do support the conclusions.

2. The statistical analysis has been performed appropriately and rigorously.

3. The authors have made all data underlying the findings in their manuscript fully available.

4. The manuscript is presented in an intelligible fashion and written in standard English.

Reviewer #2: This is a well written paper. It has addressed the objectives well. The review used a big data with many variables, which is the strength of the study. However is there any theoritical framework that support the domains used.

Reviewer #3: Review Report

Scope; Inconsistent

Abstract:Lacks clarity

-Inconsistent among the objective and the results

Background

-Not focused

-Weak

-Less detailed

-Shortened

-avoid ... from line 74

Methods

- Not objective based

-Illigiblity criteria is not intact

-The population and the outcome is not well described

-The type of studies is not presented

-Who arbitrated in case of agreement among reviewers

-Whether the retracted publications are included?

Results and discussion

-Not logical

-Lacks flow from general to specific

-Should be geraed with objective

-Is too long

-The discussion is not objective oriented, properly explained and referenced

Regards,

Reviewer #4: This paper includes all categories of SDOH as defined by the CDC, except for "health care and quality." This would be an important SDOH to look at as it is intertwined with the "inequalities" assessed in the paper. I like that the paper uses very simple methods to identify gaps in research that can be utilized and further studied through MESA. I would be curious to see how the variables in the 47 studies that combine MESA with other cohort studies differ in SDOH variables compared to the MESA-only studies (sub-analysis).

6. PLOS authors have the option to publish the peer review history of their article (what does this mean?). If published, this will include your full peer review and any attached files.

Reviewer #1: **Yes: **Kassa Demissie Abdi (PhD)

Reviewer #2: No

Reviewer #3: No

Reviewer #4: No

---

## [Author Response · Author response to Decision Letter 0]

24 Oct 2024

Reviewer #1: 

1.The manuscript is technically sound, and the data do support the conclusions.

2. The statistical analysis has been performed appropriately and rigorously.

3. The authors have made all data underlying the findings in their manuscript fully available.

4. The manuscript is presented in an intelligible fashion and written in standard English.

Response: Thank you for your review and comments.

Reviewer #2:

This is a well written paper. It has addressed the objectives well. The review used a big data with many variables, which is the strength of the study. However is there any theoritical framework that support the domains used.

Response: Thank you for your review and comment. Yes, we used the Schulz framework for Social Determinants of Health and Environmental Health Promotion, which we made more explicit in lines 95-102: “Our literature review is informed by the Schulz framework of Social Determinants of Health and Environmental Health Promotion, and thus we categorized the SSDOH according the categories outlined in the framework as follows: 1) macrosocial/structural factors (e.g., residential segregation and ideologies such as racism); 2) individual-level inequalities (e.g., employment, income, health insurance, and education); 3) built environment (e.g., land use, transportation access, and park space); 4) social context (e.g., neighborhood social cohesion and neighborhood socioeconomic status); 5) stressors (e.g., air pollution, chronic stress, violence and crime); 6) social integration and social support (e.g., social participation and social support) [13]”.

Reviewer #4: 

This paper includes all categories of SDOH as defined by the CDC, except for "health care and quality." This would be an important SDOH to look at as it is intertwined with the "inequalities" assessed in the paper. 

Response: We originally included papers that examine health insurance coverage under the “Inequalities” category. The results for those studies are reported on lines 275-276: “Consistently, access to health insurance was associated with better health outcomes [82, 195, 278, 284].” 

For clarity, we have added health insurance as an example of the inequalities category listed in the methods section: lines 95-102: “Our literature review is informed by the Schulz framework of Social Determinants of Health and Environmental Health Promotion, and thus we categorized the SSDOH according the categories outlined in the framework as follows: 1) macrosocial/structural factors (e.g., residential segregation and ideologies such as racism); 2) individual-level inequalities (e.g., employment, income, health insurance, and education); 3) built environment (e.g., land use, transportation access, and park space); 4) social context (e.g., neighborhood social cohesion and neighborhood socioeconomic status); 5) stressors (e.g., air pollution, chronic stress, violence and crime); 6) social integration and social support (e.g., social participation and social support) [13]”.

None of the studies included data on health care access, which we have now added to the results section on Built Environment (lines 319-320) as follows: “No papers focused on health care access (e.g., density of health care facilities in the neighborhood).”

I like that the paper uses very simple methods to identify gaps in research that can be utilized and further studied through MESA. 

 Response: Thank you.

I would be curious to see how the variables in the 47 studies that combine MESA with other cohort studies differ in SDOH variables compared to the MESA-only studies (sub-analysis).

Response: We have added the following text to the results section (lines 517-524): “Among these 47 papers, 85% included macrosocial measures, 15% included inequality measures, 0% included built environment measures, 11% included social context measures, 15% included stressor measures, and 6% included social support measures (percentages not mutually exclusive since some papers include more than one SSDOH). In comparison to the full sample of papers, those combining MESA data with other cohorts more often included macrosocial measures (85% vs 57%), fewer (none) included built environment measures (0% versus 14%), and fewer included stressor measures (15% versus 34%) (Table 2).”

---

## [Editor Report · Decision Letter 1]

29 Oct 2024

Structural and social determinants of health: The Multi-Ethnic Study of Atherosclerosis

PONE-D-24-30988R1

Dear Dr. Lilah M. Besser,

We’re pleased to inform you that your manuscript has been judged scientifically suitable for publication and will be formally accepted for publication once it meets all outstanding technical requirements.

Kind regards,

Andrea Da Porto

Academic Editor

PLOS ONE
---

## [Editor Report · Acceptance letter]

5 Nov 2024

PONE-D-24-30988R1 

PLOS ONE

Dear Dr. Besser, 

I'm pleased to inform you that your manuscript has been deemed suitable for publication in PLOS ONE. Congratulations! Your manuscript is now being handed over to our production team.

Kind regards, 

on behalf of

Dr. Andrea Da Porto 

Academic Editor

PLOS ONE